# Personalised pose estimation from single-plane moving fluoroscope images using deep convolutional neural networks

**Florian Vogl***, **Pascal Schütz, Barbara Postolka, Renate List, William Taylor**

Institute for Biomechanics, ETH Zürich, Zürich, Switzerland

* fv@ethz.ch

**Data Availability Statement:** The fully trained model, as well as a starting point of synthnet training are available in the ETH research repository (DOI: 10.3929/ethz-b-000544626). Access to x-ray

## Abstract

Measuring joint kinematics is a key requirement for a plethora of biomechanical research and applications. While x-ray based systems avoid the soft-tissue artefacts arising in skin-based measurement systems, extracting the object's pose (translation and rotation) from the x-ray images is a time-consuming and expensive task. Based on about 106'000 annotated images of knee implants, collected over the last decade with our moving fluoroscope during activities of daily living, we trained a deep-learning model to automatically estimate the 6D poses for the femoral and tibial implant components. By pretraining a single stage of our architecture using renderings of the implant geometries, our approach offers personalised predictions of the implant poses, even for unseen subjects. Our approach predicted the pose of both implant components better than about 0.75 mm (in-plane translation), 25 mm (out-of-plane translation), and 2° (all Euler-angle rotations) over 50% of the test samples. When evaluating over 90% of test samples, which included heavy occlusions and low contrast images, translation performance was better than 1.5 mm (in-plane) and 30 mm (out-of-plane), while rotations were predicted better than 3–4°. Importantly, this approach now allows for pose estimation in a fully automated manner.

## Introduction

Accurate measurement of joint kinematics is a key requirement for a variety of biomechanical and medical applications, such as investigating the effect of pathologies, the loading conditions and injury mechanism of joints, and the development of implants. The most common way of measuring skeletal kinematics is optical marker tracking, in which infrared cameras observe reflective markers glued to the subject's skin and determine the marker positions through triangulation. Since the markers are attached to the skin, the markers' movement differs from that of the underlying joint, thus leading to soft-tissue artefacts, which are inherent to all skin-based measurement systems [1, 2].

Contrary to such indirect approaches, x-ray methods measure the skeletal segments in a more direct manner, yielding a series of x-ray images of the joint during movement, from which the 3D kinematics can be estimated. Conventionally, this estimation has been

images (confidential, personal, human research data), and implant data (confidential data) are limited, but can be requested through contacting bt@ethz.ch under the limitations given by swiss law, the original ethics, and the non-disclosure agreements.

**Funding:** This work was supported by Innosuisse (50579.1 IP-LS, https://www.innosuisse.ch). The study sponsor had no involvement in study design, in the collection, analysis and interpretation of data, in the writing of the manuscript, and in the decision to submit the manuscript for publication.

**Competing interests:** The authors have declared that no competing interests exist.

performed by acquiring the 3D information of the target joint (e.g. using CT) and then manually adjusting the pose until the projection matches the x-ray image. This process takes about one minute per image for an experienced operator, or about five hours of manual matching work for each ten seconds of measurement. Because this process is so time-intensive, expensive, and operator-dependent [3] even for high-contrast structures such as joint replacements, there is considerable interest in a process to perform this 2D-3D pose estimation automatically.

While many approaches towards automation have been developed over recent years, the need for relatively accurate starting poses means that significant manual work is still required in practice [3–6]. With the rise of deep learning, pose estimation has attracted attention from the computer vision community, particularly for tasks such as industrial automation and robotics, and achieved impressive results in terms of performance and speed over conventional pose estimation methods. However, training deep learning methods requires large datasets with accurate pose annotations, which are extremely laborious and costly to collect for real images. Thus, most work to date has focused on everyday photographs and depth images on very specific and limited datasets [7], with only few studies exploring the medical imaging domain [8]. On top of the difficulty of acquiring a sufficient number of datasets, the strict data protection rules in many countries limit how data can be used and require special computational infrastructure to perform deep learning experiments on such data.

Over the last two decades, our research group has taken, registered, and annotated over 100'000 x-ray images of knee replacements during activities of daily living using our moving single-plane fluoroscope [9–11]. The high contrast offered by knee replacements and the standardized, high quality pose annotations of this unique dataset offers a ideal opportunity to apply deep-learning methods.

Contrary to most pose estimation tasks in computer vision, the implants' 3D information is available for every measurement—typically as a surface file from the manufacturer—allowing the pose estimation algorithm to be personalised to each subject and their specific implant combinations. While various approaches for personalising deep-learning models have been proposed, most of these methods require additional manually annotated data for the personalisation process. To avoid the need for such additional annotation, it is possible instead to employ a personalisation process based on leveraging synthethic x-ray images created by rendering the known implant geometries [12, 13].

In this work, we have taken advantage of our unique dataset and the new LeoMed ETH computational cluster, which was specifically designed to facilitate deep learning experiments on medical data, to 1) extend current deep learning approaches to pose estimation of knee replacements from x-ray images and 2) personalise these approaches to each subject and their specific implants. The vision of this work was to create a fully automatic approach to registering x-ray images in a rapid and robust manner for all radio-opaque implants.

## Materials and methods

### Dataset

Our dataset consisted of about 106'000 x-ray images with manually annotated poses, and including unique 53 tibial/femoral component combinations, where different implant sizes were considered different implants. The images were collected over the last two decades at the Institute for Biomechanics at ETH Zurich during various clinical and biomechanical studies—using our moving fluoroscope to investigate joint/implant kinematics during activities of daily living, including walking, ramp descent, and stair descent, as well as knee bending activities [9–11].

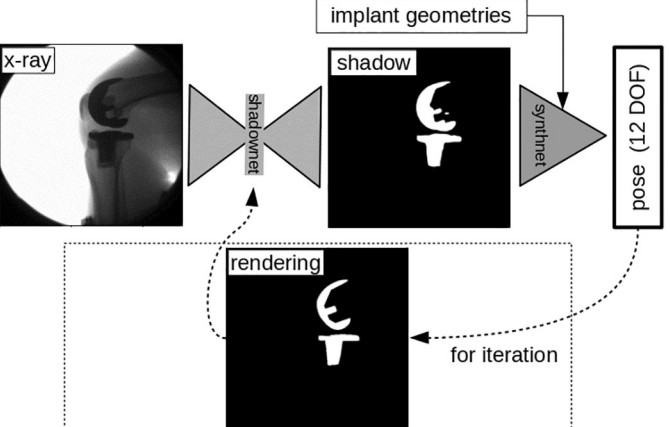

**Fig 1. The shadownet extracts the implant shadow from the x-ray image, while the synthnet estimates the pose from this shadow for the known tibia/femur implant geometries.**

For this work, the images were downscaled from 1000x1000 pixels (about 0.3 mm pixel) to 512x512 pixels to reduce memory and computational demands. Using the calibration information available for each measurement, we standardized the poses to a principal point at the center of the image intensifier and a focal distance of 0.97 m. The data was split into a training dataset from 49 of the unique tibial/femoral component combinations, and a test dataset of 4 combinations, which was used to estimate the performance of our approach on unseen subjects.

## Network architectures

The Deep Auto-Match Network (DAMN) (Fig 1) is split into two parts: a) a "shadownet" network that extracts the implant shadows from the x-ray image and b) a "synthnet" network that estimates the poses from these shadows. The intuition behind this choice is that extracting the implant shadow from the x-ray is a general process that a human could do without information on the implant geometry, just by noting the strong attenuation of the implant components and the structure of the surrounding musculoskeletal tissue, such as the thigh and shank segments. However, the same shadow could imply completely different poses for different implants, so the pose estimation step clearly needs additional information on the implants' geometries. By using rendered images of the known component geometries in varying poses, the "synthnet" learns which landmarks indicate what pose for a certain tibia/femur component combination—this process is similar to turning an unknown object to find that a certain edge is only visible for certain poses of the object. So, when this specific edge later appears in an image, the pose must be within the previously established range.

Clearly, information on the implants' geometries can help in extracting the shadow from the x-ray image—particularly for challenging images, which often include partial occlusion e.g. when the second leg passes during gait and decreases the image contrast. For such images, even a human would need additional information on how the shadow for this implant combination could appear. As a consequence, in a second experiment, we extended our architecture to iteratively consider the rendered implant shadows based on the estimated pose. From the second iteration onwards, the shadownet extracts the shadow not only from the x-ray image, but can also use the information from the rendered shadow. For the non-iterative versions of

our approach and the first iteration of the iterative version, a black image was used for this rendered shadow.

The architecture for the shadownet was based on a Deeplab-v3 segmentation network with a Resnet-50 backbone [14]. The input layer was changed to accept 2 channels—the x-ray image, and the image of the rendered components in the estimated pose. The network architecture for the synthnet was a ResNeXt-50 model [15] (32 groups of width 4) with a 1-channel input layer and a fully connected output layer giving the 18 pose regression targets—3 translation components and a 6D representation of the rotation [16] for both the tibia and femur. We chose to estimate both components at the same time to allow each component to profit from the other's likely position as well as from general information, such as features relating to surrounding structures, including the positions of the thigh and shank. We further replaced all of the batch normalisation layers with instance normalisation layers, which is better suited for the small batch sizes we use during training and our recursive architecture [17, 18].

## Synthnet-training

First, we pretrain the synthnet on synthetically rendered, binarized (shadow) images of each individual femoral/tibial implant combination. Because the parameter space of the two implant poses is so high-dimensional, using purely random poses would lead to a lot of non-sensical situations, such as when the femur and tibia are impossibly overlapping or impossibly far apart considering the anatomical constraints, or would lead to poses that are irrelevant for activities of daily living (e.g. upside-down). To keep the model focussed on the task and the computational time low, it is thus important to focus on creating implant poses that are close to the ones arising in activities of daily living. Towards this goal, we perturb poses from our dataset of activities of daily living using Gaussian noise for each of the 18 pose components to create realistic poses close to the original kinematics.

For each implant component combination, we removed all entries of this combination in the dataset for training to avoid the synthnet overfitting on any of the poses it will later encounter during the training of the shadownet. To make the synthnet more resistant to occlusions, we implemented the CutOut [19] method with a probability of 0.3, and 4 squares of 128 pixels, corresponding to one quarter of the image size. The translation components of the pose were normalized over the dataset and the rotation components were transformed from Euler angles to a 6D representation [16], giving a total of 18 regression targets.

We used batch sizes of 4 samples, and ADAM as an optimiser (learning rate of $2 * 10^{-3}$, $\beta$ values of 0.9 and 0.99, and no weight decay). As it is known that batch-size plays a key role during training [20], we accumulated the gradients of 32 samples before updating the weights during optimisation.

The pose-loss was given by a combination of translation losses $L_T$ and rotation losses $L_R$ for both the femur and the tibia poses:

$$L_{pose}(p_t, p_p) = L_{T,fem}(p_t, p_p) + L_{R,fem}(p_t, p_p) + L_{T,tib}(p_t, p_p) + L_{R,tib}(p_t, p_p) \qquad (1)$$

with $p_t \in \mathbb{R}^{18}$ being the target pose and $p_p \in \mathbb{R}^{18}$ being the predicted pose.

The translation losses $L_T$ were weighted L1-losses with the in-plane directions (x,y) having a relative weight of 10 compared to the out-of-plane direction (z). This choice reflects the fact that out-of-plane translations are more difficult to estimate, as variations in this component lead only to small changes in the size of the implant components due to perspective. A

Geodesic loss was used for the rotational loss $L_R$ [16]

$$L_R = |\cos^{-1}\left(\frac{1}{2}\left(tr(R_p R_t^T) - 1\right)\right)|, \tag{2}$$

with $R_p$ the rotation matrix corresponding to the predicted pose of either femoral or tibial implant, and $R_t$ the rotation matrix corresponding to the true pose.

Using this procedure, we first trained one synthnet to completion for about five days, and then froze the first quarter of the network before training the individualised networks for each of the 53 femoral/tibial implant combinations, which took about one day for each synthnet on a Nvidia GTX2080Ti GPU. Freezing the input layers of the synthnet ensured that the input features remained consistent between different synthnets, which was required as the synthnet part was swapped out during the training process of the shadownet in the next step.

### Shadownet-training

To train the shadownet part of the DAMN, we loaded the corresponding pretrained synthnet for each batch. By freezing the weights of the synthnet, we ensured that the training procedure only affected the weights of the generic shadownet. The optimiser was stochastic gradient descent with a learning rate of $10^{-2}$ and no momentum. Notably, we chose not to use momentum during this step, because all samples from a certain tibia/femur combination were loaded after one another. While this process avoided the requirement to frequently swap the synthnet, momentum would build up as long as the synthnet remained the same, but could be completely inappropriate as soon as the synthnet was switched to the next tibia/femur combination. The batch size for this training step was set to 2 samples for the non-iterative version and 1 sample for the iterative version of our approach. Similar to the training of the synthnet, we accumulated the gradients of 16 samples before updating the weights during stochastic gradient descent.

The loss function for this training step was given by the combination

$$L_{total}(p_t, p_p, s_p, s_t) = wL_{seg}(s_t, s_p) + L_{pose}(p_t, p_p), \tag{3}$$

of the pose loss $L_{pose}$ (Eq 1) and a Lovasz-Hinge loss, which is a smooth extension of the Jaccard index [21, 22], as the segmentation loss $L_{seg}$. Here, $p_t$ and $s_t$ are the true pose and true shadow (acquired by rendering the implants in the true pose $p_t$), while $s_p$ describes the shadow predicted by the shadownet, $p_p$ is the pose predicted by the synthnet. Further, $w = 10$ is a weight factor to make the two loss components comparable in magnitude. Training of the shadownet took about five days on a Nvidia GTX2080 Ti GPU, with improvements slowing down considerably after about three days. Overall, the training of the complete, personalised DAMN model took about 63 GPU days, which we parallelised using multiple GPUs. After the initial training, any new subject will only require a new personalised synthnet, which can be trained in 1 day using transfer-learning.

### Results

Figs 2–5 show the performance of the individualised non-iterative and iterative approaches on the testset, re-emphasizing that no data from subjects in the testset was used during training. For 50% of all images in the 4 subject test dataset, the non-iterative approach predicted the pose of both implants better than about 0.75 mm for in-plane translations and 25 mm for out-of-plane translations, while the iterative approach improved the out-of-plane predictions to below 10 mm. For 90% of the testset, the non-iterative approach achieved 1.5 mm and 30 mm (in-plane, out-of-plane) and the iterative approach achieved 2.5 mm and 25 mm. The

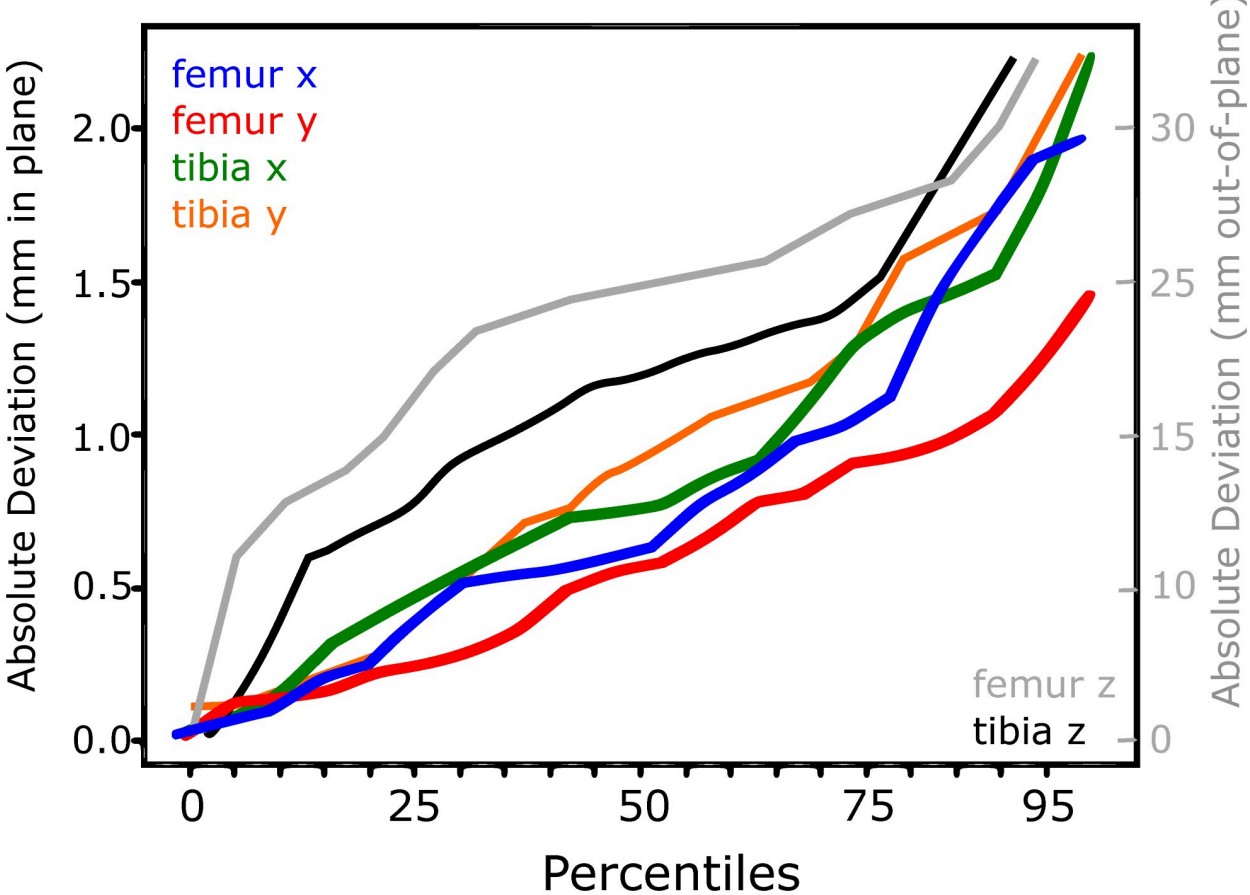

**Fig 2. Translation performance for the individualised non-iterative DAMN on the test-set.**

predicted rotations (presented as Euler angles for interpretability instead of the internal 6D representation) differed by less than about 2˚ from the ground truth over 50% of the testset for the non-iterative and less than 1.5˚ for the iterative version. Over 90% of the samples, both iterative and non-iterative versions predicted all rotations better than 3–4˚.

## Discussion

This work presented an automated method to perform 2D-3D pose estimation from fluoroscopic images based on a deep-learning approach that is individualised to each subject and their implants. For a large part of the test-set, in-plane translations were predicted to within 0.75 mm—typically for clean x-ray images with high contrast and no occlusions. Predictions were made in a fully-automated manner, in less than 10 s for 100 images, saving about 90 min required for manual annotation.

Our approach struggled for images in which the implant was occluded, as frequently happens e.g. during normal gait when the legs pass each other and decrease the contrast. Notably, such images are challenging even for humans, decreasing the quality of manual poses used as ground truths in this work. Some subjects even had two knee replacements, which can be in the image at the same time or overlap; while the algorithm can properly identify the implant to be registered (the one closer to the image intensifier), overlapping leads to a distorted shadow

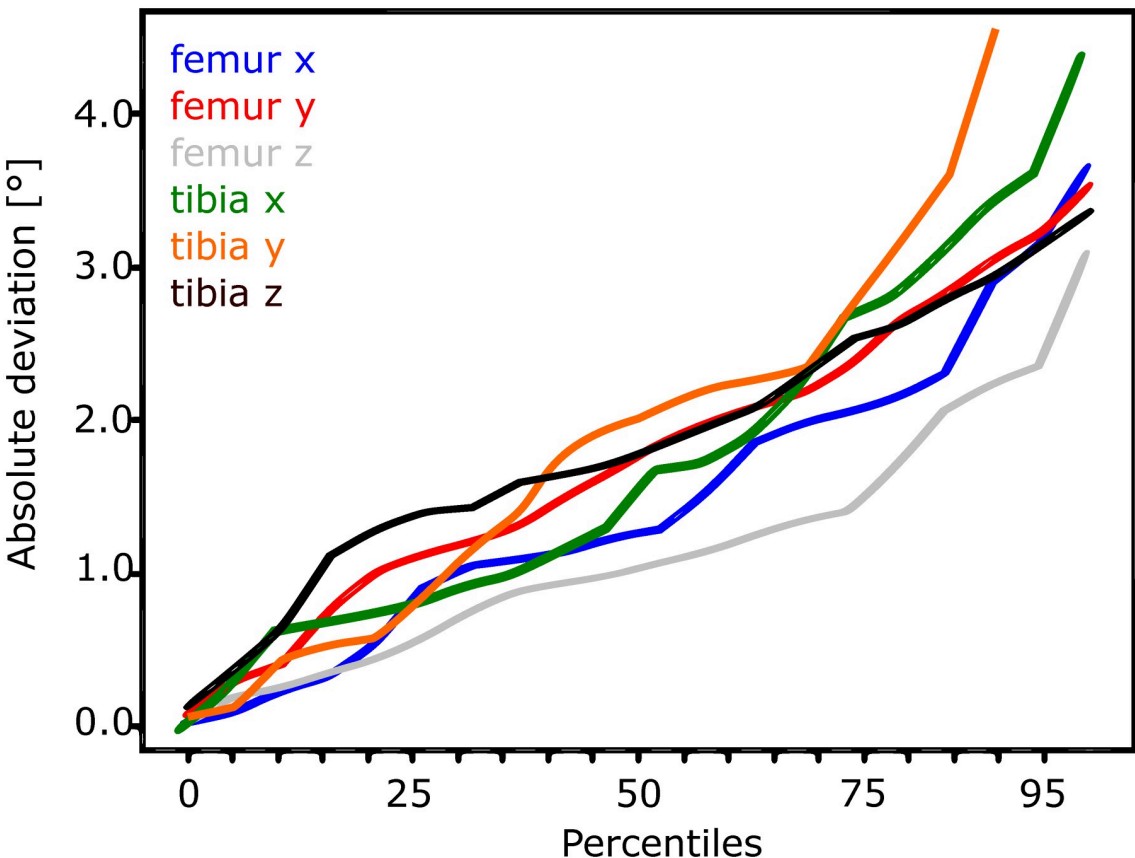

**Fig 3. Translation performance for the individualised non-iterative DAMN on the test-set.**

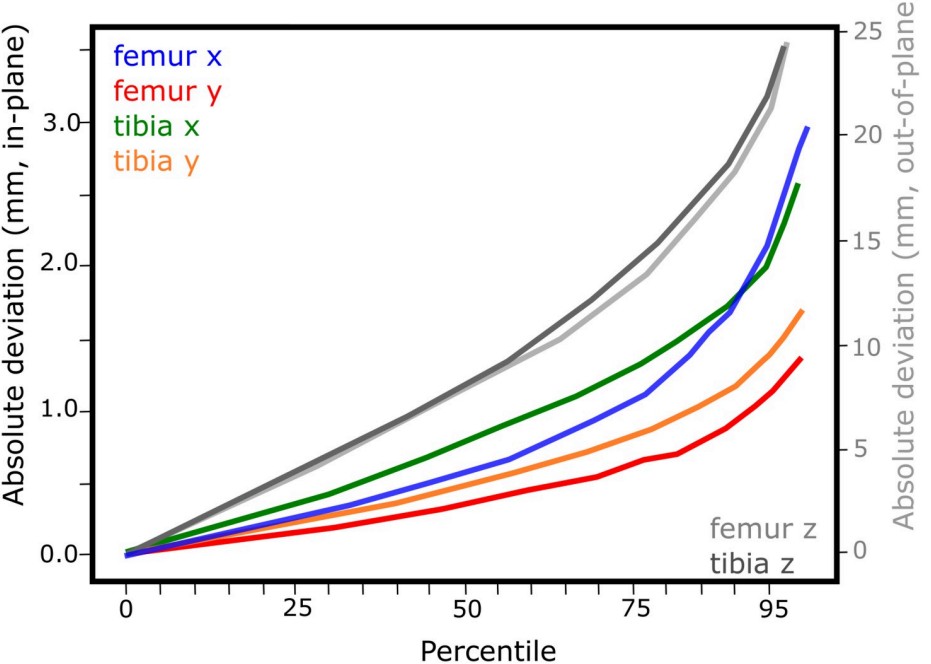

**Fig 4. Translation performance for the individualised 2-iteration DAMN on the test-set.**

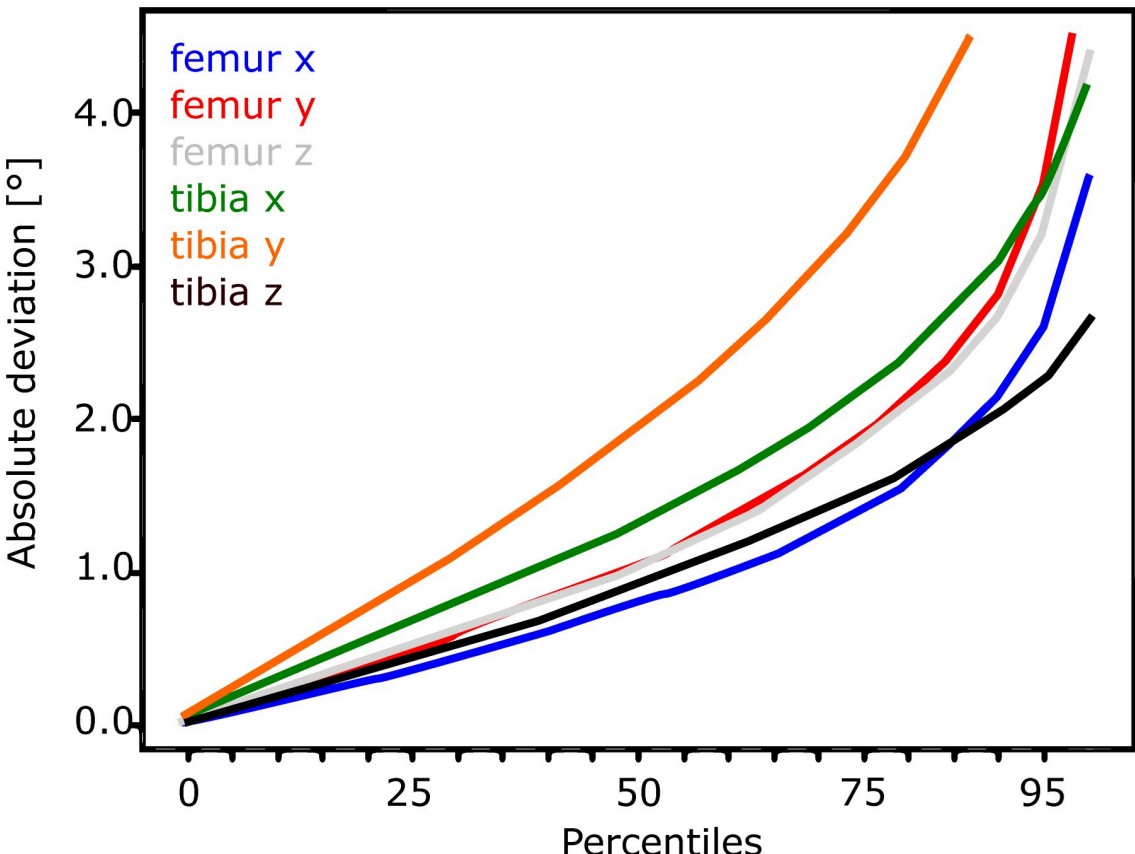

**Fig 5. Rotation (right) performance for the individualised 2-iteration DAMN on the test-set.**

segmentation. This is a classic case of distribution mismatch, because the synthnet is trained using ideal (rendered) shadows while the shadownet might not be able to provide those in all cases. For such difficult images, the in-plane translation performance dropped to about 1.5 mm. Overall, the in-plane performance is sufficiently high for most clinical and research applications on implant development and kinematics, but visual inspection of the images together with the predicted pose is advised to find major mis-predictions. Alternatively, the predicted pose could be used as the starting pose for a pose refinement optimisation [3, 4, 6], and convergence could be used as an indicator for successful pose estimation.

Even though the CutOut-method [19] used during training of the synthnet should confer some resilience to occlusions, extracting a shadow from an occluded image without any additional information simply remains challenging. Consequently, we proposed an extension to our architecture to iteratively feed the rendered implant shadows back into the shadownet, based on the estimated pose Fig 1. Suprisingly, the iterative version of our approach, which tries to provide some approximate information on how the object's full shadow could look like, did not improve predictions on such harder images, but sometimes even worsened the registration performance. The reason for this result could be that the recursive networks are harder to train and require more computational resources due to their increased depth. Even using our dedicated computational cluster, we were still only able to train the network with 1 additional iteration, which is unlikely to be sufficient for the network to learn a new iterative process—particularly for these challenging images for which the predicted shadow and pose

after the first iteration are still poor. To evaluate the true potential of the iterative approach, further investigation with a higher number of iterations will therefore be needed, most likely requiring a method to distribute the model across multiple GPUs e.g as described for volumetric data in [23].

Compared to the excellent predictions for in-plane translation, out-of-plane translations were much harder to estimate, predominantly because a change in the out-of-plane position only leads to a comparatively small change in the apparent size of the implant due to perspective. Consequently, we used a lower weight in the loss function for the out-of-plane translation, which makes the algorithm focus less on this component and resulted in a lower out-of-plane performance of 25 mm. Again, surprisingly, the iterative version of our approach improved this out-of-plane prediction to about 10 mm. As discussed above, training on networks with a higher number of iterations will be necessary to study this effect, and establish the trade-off between performance improvement and computational requirements.

Performance in pose estimation tasks is fundamentally limited by the image resolution and the quality of the 3D implant information. For example, the in-plane translation performance of our approach is about 0.75 mm, which corresponds to about one to two pixels for the downsampled 512px images. Considering this resolution limit, it is likely that the performance of our approach could be further improved by using the original 1000px images. However, use of these larger images substantially increases the size of the model and thus the required memory and training time by a factor of four. Considering the substantial training time (for a Nvidia GTX 2080 Ti: 5 days for the initial synthnet, 1 day for each implant combination, 5 days for the shadownet) this complexity increase seems infeasible impractical without making considerable improvements to either the model or the training process—particularly during development (e.g. dealing with occlusions) where rapid prototyping and evaluation is necessary.

Rather than using high resolution images in the deep-learning model directly, the predicted pose from a lower-resolution model could be used as the starting pose for a pose-refinement algorithm. Such pose-refinement algorithms have reported performances comparable to our results, when applied to high resolution images as long as they have been provided with good starting poses [3, 4, 6]. Since most of these algorithms work iteratively, one could save the one-time cost to train our deep-learning model on high-resolution images, but would pay with substantially longer times of multiple minutes per image for pose inference as compared to the mere seconds required for the deep-learning model.

One core limitation of this study is that manually annotated data are considered the ground truth. Even though a pose optimiser algorithm was applied after manual pose estimation to reduce inter-operator variability, the x-ray measurement system still had a limited resolution, and was subject to various distortion effects stemming from the experimental setup. Our results can thus only be interpreted in the context of replacing the manual pose estimation, and can only be considered as approximations with respect to the real poses. Notably, most other studies that investigated performance also suffered from this limitation, even though some aspects can be addressed through manufactured objects with known geometries [3]. On such objects, manual pose estimation achieved an in-plane performance between 0.3 mm to 0.6 mm and 7 mm out-of-plane, and predicted rotations to about 0.3 deg to 1.0 deg. However, these results are derived from relative poses, so no conclusions can be drawn about absolute poses. Furthermore, the calibration object was a simple geometric structure with good contrast, so the performance of manual pose estimation is likely to be worse for the more challenging implants, low contrast images, or images with occlusions. To assess the performance of various pose estimation approaches with respect to reality, future studies should acquire the

true pose of more realistic objects and more realistic poses through an independent and substantially more accurate means, such as mechanical precision devices.

Our approach requires that the 3D information of the target object is accurately known—if the algorithm does not know what do look for, how should it be able to find and estimate its pose? While we had no way of measuring the implant geometries directly, manufacturing discrepancies from the implant designs (which were available) are assumed to be negligible compared to other error sources. If the 3D geometry of the target object is uncertain (e.g. when estimating the pose of natural bone, which is often acquired by CT, which has a limited resolution), this effect will be more pronounced and should be investigated. Here, different discrepancies (scale, loss of detail, slight geometric changes of one part of the outline) could be investigated with respect to their effect on the estimation result and compared to the results of the gold standard using the modified geometry. While this process seems prohibitively time- and cost-intensive when using manual registration as gold standard, an automated approach using mechanical precision devices could provide a practical implementation.

For the out-of-plane translations, the information available from single-plane fluoroscopic images is inherently limited, which has led various research groups to develop dual-plane fluoroscopes [5, 24, 25]. Our pose estimation approach can directly be applied to dual-plane images by simply training the synthnet on two synthetic x-ray shadows instead of one (taking into account the relative angles and positions of the dual-plane setup). Similarly, the overall DAMN would take one additional image as an input, with no other changes required. If a trained single-plane model is already available, transfer-learning could be used to acquire a dual-plane model, requiring only minimal annotated dual-plane data.

Our approach is designed to estimate the poses frame-by-frame. While this decision allows our method to work independently of frame rate or task performed, the algorithm remains ignorant of valuable time-information (where was the object one or two frames previously or subsequently?) and the allowed movement space for a certain task (how does the joint angle typically vary with time for healthy subjects?). In principle, a variety of deep learning approaches exist to deal with time-series and video-data, but would require exorbitant computational facilities as well as training different models for each device, frame rate, or task. A recent study proposed an interesting alternative, and trained an auto-encoder to learn the possible movement space of the joint for a certain task. Then, approximate poses were improved by a Kalman-filter that considered both the time-information over multiple frames but also the kinematic understanding learned by the auto-encoder [26].

Our approach can further be extended to estimate the poses of natural bone instead of implants. Assuming that the bone geometry is known (e.g. from CT), our approach can be used with no further modifications. Indeed, for certain applications, our approach of a segmentation and a personalized regression network can also be used to estimate poses from CT-data, thus opening opportunities in other fields using CT or $\mu$-CT, such as research on porous media [27–29]. While our approach is easily extensible to deal with volumetric data (instead of 2D images) by using standard 3D-networks, the main limitation remains the requirement that the target object's geometry must be known to personalise the regression network. Because implants have high absorption and thus appear as nearly uniform shapes in the x-ray image, our approach binarizes the segmentation and regression networks. For estimating the poses of structures with lower absorption that show more internal structure on x-ray images (e.g. bone), one could replace our binary segmentation and regression networks with greyscale versions to consider this additional information. However, it is likely that this version would be more susceptible to noise or occlusion effects, so it remains to be investigated whether a binarized or grey-scale approach is advantageous for estimating the poses of natural bone.

## Conclusion

We presented a deep-learning based approach to estimate joint poses from fluoroscopic images, which is personalised to each subject without any additional manual pose registration. Depending on the required performance, our approach can be used directly to predict poses to about 0.75 mm (in-plane translation) and 2˚ (all Euler-angle rotations), and can easily be extended to natural bone, multi-plane fluoroscopy, or even CT, assuming that the target object's geometry is known. Alternatively, our approach can be combined with pose refinement algorithms [3–6] by providing approximate starting poses in a fully automated manner, thus avoiding the laborious and expensive task of manually estimating the poses for each image.

## Acknowledgments

We would like to thank Prof. Otmar Hilliges for helpful and insightful discussions.

## Author Contributions

**Conceptualization:** Florian Vogl, Pascal Schütz, Renate List, William Taylor.

**Data curation:** Florian Vogl, Pascal Schütz, Barbara Postolka, Renate List.

**Formal analysis:** Florian Vogl.

**Funding acquisition:** Florian Vogl, Pascal Schütz, William Taylor.

**Methodology:** Florian Vogl.

**Project administration:** Florian Vogl, Pascal Schütz, Renate List, William Taylor.

**Resources:** Renate List, William Taylor.

**Software:** Florian Vogl, Barbara Postolka.

**Supervision:** Florian Vogl, William Taylor.

**Validation:** Florian Vogl, Barbara Postolka, William Taylor.

**Visualization:** Florian Vogl.

**Writing – original draft:** Florian Vogl, William Taylor.

**Writing – review & editing:** Florian Vogl, William Taylor.

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
