## [Decision Letter · Decision Letter 0]

9 Mar 2022

PONE-D-21-36228Personalised pose estimation from single-plane moving fluoroscope images using deep convolutional neural networksPLOS ONE

Dear Dr. Vogl,

Thank you for submitting your manuscript to PLOS ONE. After careful consideration, we feel that it has merit but does not fully meet PLOS ONE’s publication criteria as it currently stands. Therefore, we invite you to submit a revised version of the manuscript that addresses the points raised during the review process.

We look forward to receiving your revised manuscript.

Kind regards,

Jyotismita Chaki, PhD

Academic Editor

PLOS ONE

Journal Requirements:

This work was supported by Innosuisse (50579.1 IP-LS). The study sponsor

had no involvement in study design, in the collection, analysis and interpretation

of data, in the writing of the manuscript, and in the decision to submit the

manuscript for publication. 

This work was supported by Innosuisse (50579.1 IP-LS, https://www.innosuisse.ch). The study sponsor had no involvement in study design, in the collection, analysis and interpretation of data, in the writing of the manuscript, and in the decision to submit the manuscript for publication.

6. Please remove your figures from within your manuscript file, leaving only the individual TIFF/EPS image files, uploaded separately.  These will be automatically included in the reviewers’ PDF.

Additional Editor Comments:

Kindly modify the manuscript as per the reviewer comments and resubmit

Reviewers' comments:

Reviewer's Responses to Questions

**Comments to the Author**

1. Is the manuscript technically sound, and do the data support the conclusions?

Reviewer #1: Yes

Reviewer #2: Yes

2. Has the statistical analysis been performed appropriately and rigorously? 

Reviewer #1: Yes

Reviewer #2: Yes

3. Have the authors made all data underlying the findings in their manuscript fully available?

Reviewer #1: Yes

Reviewer #2: Yes

4. Is the manuscript presented in an intelligible fashion and written in standard English?

Reviewer #1: Yes

Reviewer #2: Yes

5. Review Comments to the Author

Reviewer #1: The paper is well-motivated and the introduction of the Deep Auto-Match Network (DAMN) is novel. However, there are some concerns that should be addressed:

1. DAMN consists of two parts and contains various loss functions, and it is recommended to perform ablation experiments to show the role of each part. More explanation and intuition on DAMN are needed.

2. It is recommended that the time complexity should be discussed and analyzed in detail. In addition, the parameters of the DAMN should also be analyzed numerically.

3. Since some data is difficult to obtain, it is worth considering whether the performance can be improved by generating data. For example, some work similar to [1] may have some inspiration.

[1] Gong, An, Xinjie Yao, and Wei Lin. "Dermoscopy image classification based on StyleGANs and decision fusion." Ieee Access 8 (2020): 70640-70650.

4. There are several works on the combination of deep learning and medicine, and it is suggested to compare with some state-of-the-art methods. In addition, this work discussed x-ray CT and its strengths and disadvantages, it is time-intensive, expensive, and operator-dependent. In fact, in the field of porous media/materials, the same problem exists with the use of X-CT, see, Fractals-Complex geometry patterns and scaling in nature and society, 2018, 26(2): 1840003. https://doi.org/10.1142/S0218348X18400030, and Physics of Fluids, 2021, 33(3): 032013. https://doi.org/10.1063/5.0042606. It is suggested that the authors refer to these recent works to strengthen the technical context of this ms, support the point of view about X-CT and attract more readers.

Reviewer #2: Paper deals with hot topic. It has scientific novelty and great practical value.

Paper has a logical structure all necessary sections. The paper is technically sound. The experimental section is good.

The proposed approach is logical, results are clear.

Suggestions:

(1) Can authors explain more about 6D representation of predicted rotation and automated method to perform 2D-3D pose estimation?

(2) How the quality of the 3D implant information effects performance in pose estimation task? Justify.

(3) In common we uses high resolution images in deep learning algorithms. What is your approach? If you are not using high resolution images then why?

(4) If it is not restricted in journal norms then one Conclusion Section will help interested authors to attract readers to go read this research work.

6. PLOS authors have the option to publish the peer review history of their article (what does this mean?). If published, this will include your full peer review and any attached files.

Reviewer #1: No

Reviewer #2: No

---

## [Author Response · Author response to Decision Letter 0]

2 May 2022

We would like to thank the reviewers for their critical feedback, which we have included and have helped to improve our manuscript. Please see the attached file "Response to Reviewers" for a detailed point-by-point response, including the relevant modifications for your convenience.

---

## [Decision Letter · Decision Letter 1]

14 Jun 2022

Personalised pose estimation from single-plane moving fluoroscope images using deep convolutional neural networks

PONE-D-21-36228R1

Dear Dr. Vogl,

We’re pleased to inform you that your manuscript has been judged scientifically suitable for publication and will be formally accepted for publication once it meets all outstanding technical requirements.

Kind regards,

Jyotismita Chaki, PhD

Academic Editor

PLOS ONE

Additional Editor Comments (optional):

I am happy to inform you that reviewers are satisfied with the revised manuscript and thus I am provisionally accepting the manuscript for publication.

Reviewers' comments:

Reviewer's Responses to Questions

**Comments to the Author**

1. If the authors have adequately addressed your comments raised in a previous round of review and you feel that this manuscript is now acceptable for publication, you may indicate that here to bypass the “Comments to the Author” section, enter your conflict of interest statement in the “Confidential to Editor” section, and submit your "Accept" recommendation.

Reviewer #1: All comments have been addressed

2. Is the manuscript technically sound, and do the data support the conclusions?

Reviewer #1: Yes

3. Has the statistical analysis been performed appropriately and rigorously? 

Reviewer #1: Yes

4. Have the authors made all data underlying the findings in their manuscript fully available?

Reviewer #1: Yes

5. Is the manuscript presented in an intelligible fashion and written in standard English?

Reviewer #1: Yes

6. Review Comments to the Author

Reviewer #1: (No Response)

7. PLOS authors have the option to publish the peer review history of their article (what does this mean?). If published, this will include your full peer review and any attached files.

Reviewer #1: No

---

## [Editor Report · Acceptance letter]

16 Jun 2022

PONE-D-21-36228R1 

Personalised pose estimation from single-plane moving fluoroscope images using deep convolutional neural networks 

Dear Dr. Vogl:

I'm pleased to inform you that your manuscript has been deemed suitable for publication in PLOS ONE. Congratulations! Your manuscript is now with our production department. 

Kind regards, 

on behalf of

Dr. Jyotismita Chaki 

Academic Editor

PLOS ONE